

# The photo-inhibition of camphor leaves (*Cinnamomum camphora* L.) by NaCl stress based on physiological, chloroplast structure and comparative proteomic analysis

Jiammin Yue[1,2,3], Dawei Shi[1], Liang Zhang[1], Zihan Zhang[1], Zhiyuan Fu[1], Qiong Ren[4] and Jinchi Zhang[1]

[1] Co–Innovation Center for Sustainable Forestry in Southern China of Jiangsu Province & Key Laboratory of Soil and Water Conservation and Ecological Restoration, Nanjing Forestry University, Nanjing, Jiangsu, China
[2] Key Laboratory of Land Degradation and Ecosystem Restoration & Key Laboratory of Rehabilitation and Reconstruction of Degraded Ecosystems in Northwest China, Ningxia University, Yingchuan, Ningxia, China
[3] Department of Forest and Conservation Sciences, Faculty of Forestry, University of British Columbia, Vancouver, British Columbia, Canada
[4] Jiangxi Academy of Forestry, Nanchang, China

Corresponding author
Jinchi Zhang,
zhangjinchi8811@163.com

## ABSTRACT

**Background**. The distribution and use of camphor (*Cinnamomum camphora* L.) trees are constrained by increasing soil salinity in south-eastern China along the Yangtze River. However, the response mechanism of this species to salinity, especially in team of photosynthesis, are unknown.

**Methods**. Here, we analysed themorphological, physiological, ultrastructural, and proteomic traits of camphor seedlings under NaCl (103.45 mM) treatment in pot experiments for 80 days.

**Results**. The growth was limited because of photosynthetic inhibition, with the most significant disturbance occurring within 50 days. Salinity caused severe reductions in the leaf photosynthetic rate ($A_n$), stomatal conductance ($g_s$), maximal chlorophyll fluorescence ($F_m$), maximum quantum yield of PSII ($F_v/F_m$), non-photochemical quenching (NPQ), relative quantum efficiency of PSII photochemistry ($\Phi$PSII), photochemical quenching coefficient (qP) and photo-pigment contents (chlorophyll a (Cha), chlorophyll b (Chb), total chlorophyll (Chl)); weakened the antioxidant effects, including those of malondialdehyde (MDA), superoxide dismutase (SOD) and peroxidase (POD); and injured chloroplasts. The physiologicalresults indicated that the main reason for photo-inhibition was oxidative factors induced by NaCl. The proteomic results based on isobaric tags for relative and absolute quantitation (iTRAQ) further confirmedthat photosynthesis was the most significant disrupted process by salinity ($P < 0.01$) and there were 30 downregulated differentially expression proteins (DEPs) and one upregulated DEP related to restraint of the photosynthetic system, which affected photosystem I, photosystem II, the Cytochrome b6/f complex, ATP synthase and the light-harvesting chlorophyll protein complex. In addition, 57 DEPs were related to photo-inhibition by redox effect and 6 downregulated DEPs, including O2 evolving complex 33kD family protein (gi|224094610) and five other predicted

proteins (gi|743921083, gi|743840443, gi|743885735, gi|743810316 and gi|743881832) were directly affected. This study provides new proteomic information and explains the possible mechanisms of photo-inhibition caused by salinity on *C. camphor*.

# INTRODUCTION

Global salinization in agroforestry lands occurs due to natural processes and human interventions (*Zhang & Yu, 2019*). The amount of primary salt-affected soil worldwide is approximately 955 M ha, while secondary salinization affects some 77 M ha of land area (*Metternicht & Zinck, 2003*). Salinity causes water deficits, ion toxicity/imbalance, and oxidative stress in plants and damage to cells and organs, greatly limits vegetative growth. For example, 50~200 mM NaCl induced the photosynthetic ability reduction, photo-pigments decrease, the disturbance of elements and ions content, increase of malondialdehyde (MDA) and activation of some antioxidant enzymes (SOD, CAT, APX) in different plants such as *Capsicum annuum* (*Manivannan et al., 2016*), *Zelkova serrate* (*Wang et al., 2019*), *Larix decidua* Mill. (*Plesa et al., 2018*) and *Lonicera japonica* L. (*Zhao et al., 2015*), which all cause the growth limitation to varying degrees and may even cause death.

Photosynthesis plays a pivotal role in plant life cycles; however, it is highly affected by salt stress. Under salinity, plants exhibit a distinct physiological and biochemical profile of photosynthesis, as increased salinity results in the downregulation of some photosynthetic genes (*Chaves, Flexas & Pinheiro, 2009*). PS1 and PS2 are two-electron transport systems that are involved in photosynthesis and contain pigment-protein complexes. The expression of genes controlling chlorophyll, which participates in the bioprocessing of PSI and PSII, can be inhibited by salinity; thus, plant photosynthesis can be affected (*Sudhir & Murthy, 2004*). Salinity inhibits the synthesis of the psbA, ATP synthase beta, ATP synthase alpha and CP47 protein, causes harm to chloroplasts and elevates the contents of $Na^+$, $Cl^-$ and $H_2O_2$ in cucumber (*Cucumis sativus* L. cv. Jinyou No. 4) (*Shu et al., 2015*); the LHCSR1 protein, a member of the light-harvesting complex (LHC) family, active in non-photochemical quenching (NPQ) in moss (*Physcomitrella patens*), folds correctly only with chlorophyll a (Chl a) and xanthophylls (*Pinnola et al., 2015*), which results in the disturbance on the accumulation of Chl a caused by salinity and ultimately inhibits energy transformation in the photosynthesis process.

The evergreen camphor tree (*Cinnamomum camphora* L.) is a native subtropical species. It has been universally planted in southern China, especially along the Yangtze River valley (including Jiangsu province) (*Chen et al., 2010*). As a most important economic species for thousands of years in this area, camphor has been well known for its properties as a natural repellent, as well for its as timber and ornament value (*Zhou & Yan, 2016*). In addition, it is now utilized as a medicinal plant for the treatment of muscular strains,

inflammation, rheumatic conditions and is considered an essential oil-producing tree (*Shi et al., 2009*). As a habitat for camphor, Jiangsu is climatically situated within a transitional region between the warm temperate zone and the subtropical zone in China (*Zhenzhen et al., 2009*; *Pan, Li & Zhang, 2005*). In the context of global warming, sea levels along the shore are rising. Further, evaporation and the capillary rise of saline groundwater will lead to NaCl accumulations on the surface of saline soils. A total land area of 15 km$^2$ is degraded annually along the coast of Jiangsu (*Chen, Shen & Xu, 2007*). Moreover, younger trees are less salt-tolerant than older trees (*Munns & Tester, 2008*; *Wang et al., 2013*). Consequently, camphor seedlings in nursery gardens and in their natural habitat are exposed to higher NaCl levels compared with those of mature plants.

Camphor is sensitive to NaCl (*Gilman & Watson, 2016*). A 20-day treatment with 200 mM Na$^+$ resulted in decreased pigment contents, antioxidant enzyme activities and photosynthetic rate ($A_n$) in camphor leaves (*Han et al., 2014*). However, the response mechanisms of camphor seedlings to salinity disturbance have been studied little; hence, an improved understanding of the bio-physiological variation induced by NaCl, especially in relation to the photosynthetic process, could facilitate the identification of measures involved in improving salinity adaption.

Isobaric tags for relative and absolute quantitation (iTRAQ) analysis has been used for elucidating various mechanisms of responses to salt stress radish (*Raphanus sativus* L.) (*Sun et al., 2017*), *Paulownia fortunei* (*Deng et al., 2017*), tomato (*Lycopersicon esculentum*) and apple (*Malus domestica*) (*Hu et al., 2016*). The current study attempts to investigate the influence of 103.45 mM NaCl, which is the average salinity of Jiangsu coastal lands, on camphor seedling by the iTRAQ method combined with physiological analyses. With the main hypothesis that camphor growth in salinity will be limited due to photosynthetic limitation, our present work aims to (i) explore the variability of physiological responses of camphor under both control and 103.45 mM NaCl and (ii) attempt to both identify the important parameters and determine the key proteins that possibly affect photosynthetic limitations.

## MATERIALS & METHODS

### Plant materials and salt stress treatment

The seeds of the Camphor tree (*Cinnamomum camphora* L.) were collected from Lianxi region, Jiujiang City, Jiangxi province in China. All the seeds were cleaned with soap and rinsed with purified water, then soaked in 30% H$_2$O$_2$ solution for 30 min at 22 °C. Next, the seeds were washed under flowing distilled water for 24 h. Seeds were soaked for further 36 h at room temperature (20 °C) in distilled water. After these procedures, all the seeds were sown into plates containing quartz sand, which kept the material humidity to field capacity of ∼35% for germination. The temperature was fixed at 5 °C. Hypocotyls appeared after approximately 2 months, at which time the germinant seeds were transplanted into seedling bags (10 cm diameter × 15 cm depth) filled with a mixture of vermiculite, peat and perlite (1:2:2, v/v).

Seedlings were cultivated under natural light in a greenhouse at Nanjing Forestry University (32°7′N, 119°12′E), Jiangsu, China. The day/night temperature regimes were

at 26 °C/16 °C and seedlings were watered regularly with tap water. The seedlings were watered with modified Hoagland's solution every 25 days, following the method described by *Hoagland & Arnon (1950)*. Hoagland's solution comprises the following: 0.2 mM $KH_2PO_4$, 1.0 mM $K_2SO_4$, 2.0 mM $Ca(NO_3)_2$, 0.5 mM $MgSO_4$, 200 μM Fe-EDTA, 5 μM $H_3BO_3$, 2 μM $MnSO_4$, 0.5 μM $ZnSO_4$, 0.3 μM $CuSO_4$, and 0.01 μM $(NH_4)_2Mo_7O_{24}$; pH = 7.0). After 3 months of germination, seedlings reached approximately 30 cm in height. They were then transplanted into plastic pots (40 cm × 25 cm × 15 cm; 1 plant per pot) containing 3.5 kg of coarse sand and vermiculite 2:1 (v:v). One month later, the seedlings that reached approximately 40 cm in height were selected for experiment.

The selected seedlings were separated into two groups, which were both watered weekly for seven weeks: one was given water with no NaCl solution added (CK), and the other was given 0.5 L of 103.45 mM NaCl solution weekly, with salinity concentration increased gradually to end at 103.45 mM of the material in plastic pots (3.5 kg). The first fluorescence, photosynthetic indexes, pigments, malondialdehyde and antioxidant enzyme measurement of the plants took place on the 30th day (i.e., 31 weeks after germination), then on the 50th day, and finally on the 80th day after NaCl treatment. Other indexes were measured around the 50th day of NaCl treatment. Over the entire measurement period, the day/night temperature regimes were at 29 °C ±2.6/19 °C ±1.9, with relative humidity varying from 45–80%, and natural lighting provided a mid-day photosynthetic photon flux density (PPFD) of ~1,000 μmol m$^{-2}$ s$^{-1}$.

## Physiological index and ultrastructure measurements
### Measurement of plant growth
Seedling height, diameter and leaf area were determined with a ruler, digital calipers and a CI-203 portable laser area meter (CID Inc., Camas, USA), respectively, on the day before salt stress treatments were initiated, and again after 50 days.

### Photosynthetic indexes
Using an ambient $CO_2$ concentration ($C_a$) of 378 ± 33.4(SD) μmol mol$^{-1}$ by infrared gas exchange analyzer (CIRAS-2 portable photosynthesis system, Amesbury, MA, USA), leaf gas exchange indexes such as photosynthetic rate ($A_n$), stomatal conductance ($g_s$), and intercellular $CO_2$ concentration ($C_i$), were measured in leaf area of 0.7 cm × 3.0 cm (2.1 cm$^2$). This measurement took place between 09:30 am and 11:30 am local time in November on clear days when relative air humidity and leaf temperature in the chamber were maintained at ~35% and 25 °C, respectively. PPFD ranged from 1,200 to 1,400 μmol m$^{-2}$ s$^{-1}$. In this analysis, four fully expanded leaves of each seedling group were assayed. The ratio of $C_i$ to $C_a$ was calculated as $C_i/C_a$, and the intrinsic water use efficiency ($WUE_i$) was calculated as $WUE_i = A_n /g_s$ (*Li et al., 2013*).

### Chlorophyll fluorescence indexes
Parameters were measured on the leaves using a portable fluorometer (PAM-2500; Walz, Effeltrich, Germany) as employed in *Kalaji et al. (2011)*. The leaves were tested for minimal ($F_o$) and maximal ($F_m$) chlorophyll fluorescence emission after they were kept in dark for 30 min prior to measuring. The minimal ($F_o$') and maximal ($F_m$') fluorescence level

in the light-adapted state and the steady-state value of fluorescence ($F_s$) were measured after light adaption at 600 $\mu$mol m$^{-2}$s$^{-1}$, then photochemical quenching coefficient (qp) was measured. The maximum quantum yield PSII ($F_v/F_m$), thermal dissipation ($H_d$), non-photochemical quenching (NPQ), relative quantum efficiency of PSII photochemistry ($\Phi$PSII) were defined as $F_v/F_m$, 1−($F_v'/F_m'$), ($F_m$−$F_m'$)/$F_m'$, ($F_m'$−$F_s$)/$F_m'$ respectively.

### Chloroplast ultrastructure observation

The ultrastructure of the chloroplasts was characterized as follows: The leaf segments were cut into pieces that were approximately 1 mm$^2$ in size, then immersed in a 0.1 M phosphate buffer (pH = 7.4) with 3% glutaraldehyde and 1% formaldehyde for 2 h to finish the primary fixation. Then, 2% osmic acid was added in the same buffer to accomplish the secondary fixation for another 2 h. After dehydration in acetone and embedding in Durcupan ACM (Fluka), ultrathin sections were cut, stained with uranium acetate and lead citrate in sequence, then examined using a Hitachi transmission electron microscope (Carl Zeiss, Göttingen, Germany) at an 80 kV acceleration voltage (*Shu et al., 2013*).

### Assays for photosynthetic pigments

0.1 g freshwater weight (FW) of leaves for salt treatments from the samples of laminal tissue, respectively, were ground using a mortar and pestle. Then, ice-cold, 95% (v/v) ethanol was used to extract total chlorophyll (Chl), chlorophyll a (Chl a), chlorophyll b (Chl b), and total carotenoids (Car). The extracts were centrifuged at 6,000 g for 10 min, and the supernatant was collected. The precipitate was again washed with 95% (v/v) ethanol and centrifuged in order to collect the supernatant, which was combined with the previously collected supernatant mentioned above. The absorbance of the combined supernatant was read spectrophotometrically and calculated by reading the absorbance at 470, 663 and 645 nm and calculating the value of Chl a/Chl b (*Wang, 2009*; *Yao et al., 2012*).

### Assays for malondialdehyde and antioxidant enzyme activities

200 mg of fresh leaf samples of salt treatments taken from the plant were pulverized with a mortar and pestle in liquid nitrogen before adding 0.05 M phosphate buffer (pH = 7.0) for further grinding. Then, the suspension was collected in a test tube and diluted with the same buffer to 10 mL for testing enzyme activity by He's research (*He et al., 2016*). Superoxide dismutase (SOD) was assayed with the NBT method (*Fridovich, 1975*) and expressed in terms of units min$^{-1}$ g$^{-1}$ FW. One unit (U) of SOD was defined as the amount of enzyme required to cause 50% inhibition of the reduction of NBT as monitored at 560 nm. Peroxidase (POD) was measured by monitoring the rate of guaiacol oxidation at 470 nm (*Hammerschmidt, Nuckles & Kuć, 1982*). The standard curve was constructed using 4-methoxyphenol and activity was expressed as $\mu$g oxidized min$^{-1}$ g$^{-1}$ FW. Lipid peroxidation was determined by measuring the amount of malondialdehyde (MDA) produced per g FW by the thiobarbituric acid reaction as described in the paper of *Xu et al. (2018)*.

## Protein extraction, labeling and analysis
### Protein extraction

This procedure was done by placing 0.8 g of frozen leaf samples into a cold mortar. Four different treatments were prepared in total, including two other treatments (not contained in this research), one with 103.45 mM NaCl, and one without any treatments. 0.1 g leaf sample was used in each group from the 50d salinity treatment and they were duplicated. In this research, the data came from just two treatments: non-salinity treatment (C) and 103.45 mM NaCl treatment (S), the aim of which was to clarify the influence on camphor seedlings of NaCl alone. The tissues were first ground into fine powder using a mortar and pestle. The powder was suspended in 1 mL of phenol extraction buffer including 0.7 M of sucrose, 0.1 M of NaCl, 0.5 M of Tris-HCL (pH = 7.5), 50 mM of EDTA-2Na, and 0.2% of DTT. The mixture was incubated at room temperature for 10 min. Afterwards, another 1 mL of phenol saturated with Tris–HCl (0.1 M Tris, pH = 8.0) was added and the mixture was vortexed and incubated for 40 min at 4 °C. Total protein was recovered in the supernatant by centrifugation at 5,000 rpm for 15 min at 4 °C. Then 0.1 M of cold ammonium acetate-methanol solution (Using five-fold volumes of the collected phenolic phase) was added and incubated for 12 h at −20 °C until obvious delamination appeared. The sediment was separated by centrifugation at 12,000 rpm for 10 min at 4 °C and washed twice with cold methanol, then mixed with acetone at a ratio of 1 to 5 in volume by centrifuging. After drying at room temperature for 2 min, the pellet was re-suspended in 300 μL lysate solution (SDS) lysis buffer (beyotime, China, NO. P0013G) for 3 h at room temperature. This step was repeated twice. The precipitate was removed by centrifugation at 12,000 rpm for 10 min at room temperature and the supernatant was collected. The supernatant containing extracted proteins was used for further analysis. The concentrations of the protein extracts were measured using the BCA method (*Smith et al., 1985*). Samples of 30 μg for the 12% SDS-PAGE gel were laid, and the gel was visualized by Coomassie brilliant blue (CBB) stain according to Candiano's method (*Candiano et al., 2004*). After that, the stained gel at a resolution of 300 dots per inch was scanned using the Image Scanner (GE Healthcare, USA).

### Digestion and iTRAQ Labeling

The following method was used in accordance with the method of Wiśniewski's team (*Wiśniewski et al., 2009*). 100 μg of protein extract was suspended in 120 μL of reducing buffer that consists of 10 mM of DL-Dithiothreitol (DTT), 8 M of Urea, 100 mM of Tetraethyl-ammonium bromide (TEAB, pH = 8.0). The suspension was incubated for 1 h at 60 °C Iodacetamide (IAA) was then added until the final concentration was 50 mM in a dark setting, and it was incubated for 40 min at room temperature. The bottom part of the solution was collected by centrifugation at 12,000 rpm for 20 min and washed twice with 100 μL 100 mM of TEAB by centrifuging at 12,000 rpm for another 20 min. The filter units were mixed with 100 μL 100 mM of TEAB and 2 μL of sequencing-grade trypsin (1 μg/μL) and incubated at 37 °C for 12 h. Subsequently, peptide was recovered by centrifuging at 12,000 rpm for 20 min. The collected solution was combined with lyophilize. Afterwards, the sample was reconstituted in 100 μL 100 mM of TEAB, 40 μL of which was used for

labeling by iTRAQ kits. 200 µL of isopropanol was mixed with the sample and vortexed. 100 µL of iTRAQ reagent was mixed with samples and incubated at room temperature for 2 h. The labeling reaction was stopped by addition of 200 µL of distilled water. Samples labeled with the iTRAQ tags were defined as sample "C" (113,114 tags) and sample "S" (115,116 tags).

### Mass spectrometry analysis

The sample was lyophilized and stored at $-80$ °C before mass spectrum analysis. All analyses were performed by a Triple TOF 5600 mass spectrometer (SCIEX, USA) equipped with a Nanospray III source (SCIEX, USA). Samples were loaded onto a capillary C18 trap column (3 cm × 100 µm) and separated by a C18 column (15 cm × 75 µm) on an Eksigent nanoLC-1D plus system (SCIEX, USA). The flow rate was 300 nL/min and linear gradient was 90 min (from 5–85% B over 67 min; mobile phase A = 2% acetonitrile /0.1% formic acid and B = 95% acetonitrile / 0.1% formic acid). Other operation conditions were 2.4 kV ion spray voltage, 35 psi curtain gas, 5 psi nebulizer gas, and an interface heater temperature at 150 °C. The mass spectrograph (MS) was scanned between 400 and 1,500 m/z (1 Megaword with a mass-to-charge ratio) with an accumulation time of 250 ms. For Information Dependent Analysis (IDA), 30 MS/MS spectra (80 ms each, mass range 100–1,500 m/z) of MS peaks above intensity 260 as well as a charge state of between 2 and 5 were implemented. A rolling collision energy voltage was used for CID fragmentation for MS/MS spectra acquisitions. The mass was dynamically excluded for 22 s.

### Database search and protein quantification

ProteinPilot software (version 4.0, AB Sciex Inc., USA) was used to perform the relative quantification of protein from NCBI proteomic database on *Populus euphratica*. To reduce the false identification of peptides, only the peptides with 95% confidence interval were used. The standard for following the screening criteria of reliable proteins was to dismiss >1.3 unused and ≥ 1 unique peptides to remove invalid values and the anti-library data. Then the significantly differentially expressed proteins (DEPs) were screened based on the reliable proteins. The protein ratio (fold-change) was according to the group (S/C). Two criteria were to screen the DEPs: (1) a protein must contain at least two unique spectra, and only these unique spectra are for the quantification of proteins; (2) the protein ratio standard had to have a fold-change ≥2.0 or ≤0.5. The proteins that met both criteria were considered as the differential proteins (*Yan et al., 2017*).

### Bioinformatics analysis of differential proteins

After carrying out the quality control of raw data by a hierarchical cluster analysis algorithm (*Yan et al., 2017*), all the credible and different proteins were carried on to the Gene Ontology (GO) analysis based on DAVID 6.7 and the pathway analysis by the Kyoto Encyclopedia of Genes and Genome (KEGG) database (*Kanehisa & Goto, 2000*).

## Statistical analysis

The proteomic analysis had two replicates and the physiological analyses had at least three replicates. Two-way analyses of variance (ANOVA) were employed to test the effects of salt

stress, the time of salinity treatments and their interaction by R 3.3.0 (SPSS Inc., Chicago, IL, USA) and means were separated using Duncan's multiple range tests. Where necessary ($A_n$, $C_i/C_a$, MDA, POD and SOD data were square root and/or log transformed to meet assumptions of normality and homogeneity of variance. Standard error bars in figures were based on untransformed data. Differences were considered significant at $P < 0.05$. All the data met the assumptions of normality and homogeneity of variance. Origin 8.0 was used to make the charts.

## RESULTS

### Effect of salt stress on plant growth, physiological indexes and chloroplast ultrastructure

The 103.45 mM NaCl induced a severe reduction in growth in terms of seedling height, diameter and leaf area ($\Delta$Height, $\Delta$Diameter, $\Delta$Leaf area) (Table 1). The salinity caused an obvious decrease in photosynthesis rate ($A_n$) and stomatal conductance ($g_s$) and an increase in intrinsic water use efficiency ($WUE_i$) at each time point ($P < 0.01$). The reduction in $A_n$ and $g_s$ was not different between 50 days and 80 days, and the enhancements of $WUE_i$ were not different in either period caused by salinity. The significant reduction in the ratio of intercellular to ambient $CO_2$ concentration ($C_i/C_a$) appeared only at 30 days, and the treatment effects of NaCl on 50 and 80 days on $C_i/C_a$ were not distinguished from those of the non-salinity treatments. The two-way analyses of variance (ANOVA) showed the difference of photosynthetic indexes ($A_n$, $C_i/C_a$, $g_s$, $WUE_i$) among NaCl stress (N) with $P < 0.01$. The time of NaCl treatments (T) induced the effect in $A_n$, $C_i/C_a$, $g_s$ with $P < 0.05$ and $WUE_i$ with $P = 0.7807$. The interactive statistical effects in NaCl treatments and NaCl stress (T × N) of $A_n$, $C_i/C_a$, $g_s$ $WUE_i$ were $P < 0.01$, $P = 0.325$, $P < 0.01$ and $P = 0.9982$ (Fig. 1 and Table 2). Salinity led to a significant decrease in the maximal chlorophyll fluorescence emission ($F_m$), maximum quantum yield of PSII ($F_v/F_m$), relative quantum efficiency of PSII photochemistry ($\Phi$PSII) and photochemical quenching coefficient (qP) compared to those in the CK at 30, 50 and 80 days. NaCl caused an obvious reduction only in non-photochemical quenching (NPQ) and an improvement in thermal dissipation ($H_d$) after 50 days. The ANOVA analysis described the difference of fluorescence indexes ($F_m$, $F_v/F_m$, $\Phi$PSII, qP, NPQ, $H_d$) among N with $P < 0.01$. The T caused the influence in $F_m$, $F_v/F_m$, $\Phi$PSII with the $P < 0.01$ and $H_d$ with $P = 0.0110$. $F_v/F_m$, NPQ, $H_d$ responded to T × N by $P < 0.01$, $P = 0.0279$ and $P = 0.0263$. Besides, the influence of above other indexes ($F_m$, $\Phi$PSII, qP) to T and T × N was $P > 0.05$ (Fig. 2 and Table 2).

The photosynthetic pigments (Chl a, Chl b, their ratios) (Chl a/ Chl b and total Chl) had an obvious decrease from 30 days to 80 days. In addition, for Chl a/Chl b, a significant reduction occurred in 80 days. At 30 days, the Chl a/Chl b showed only a small increase, which was not significant. The reduction in Chl a was more severe than that in the others. The difference of pigments content (Chl a, Chl b and Chl) among N with P<0.01 except Chl a/ Chl b ($P = 0.0128$). The effect in Chl b and Chl among T treatment showed $P = 0.0176$ and $P = 0.0126$. The role of T × N in Chl a/Chl b and Chl were $P = 0.0494$ and $P = 0.0345$. However, the responds of other pigments (Chl a, Chl b) to T and T × N showed $P > 0.05$

**Table 1  Effects of salt stress on growth indexes.** Effect of salt stress on increments in seedling height growth, diameter growth, and leaf area growth of camphor seedlings over a period of 50 days. Initial height, diameter and leaf area were $39.4 \pm 0.9$ cm, $4.0 \pm 0.3$ mm, and $15.1 \pm 1.0$ cm², respectively.

| Treatments | Δ Height cm | Δ Diameter mm | Δ Leaf area cm² |
|---|---|---|---|
| CK | $18.1 \pm 0.5$ | $3.1 \pm 0.1$ | $8.8 \pm 1.8$ |
| 103.45 mM NaCl | $6.8 \pm 0.6$[**] | $0.4 \pm 0$[**] | $1.1 \pm 0.1$[**] |

**Notes.**
[**] Significant differences between non-salinity (CK) and salinity stress (103.45 mM NaCl) treatments ($P < 0.01$).

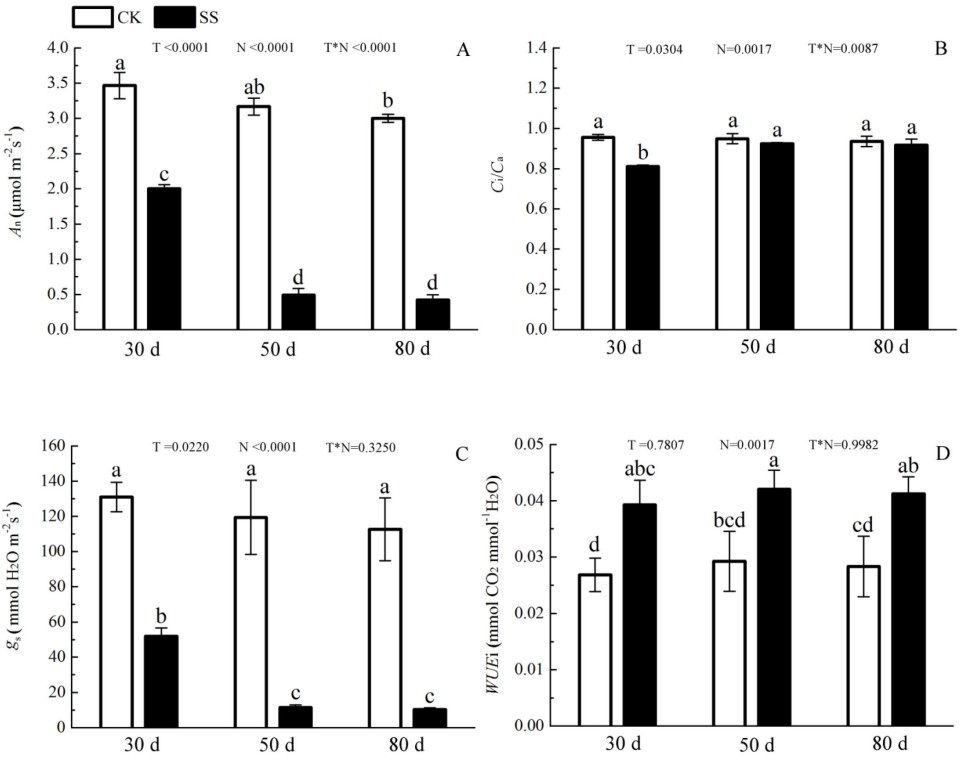

**Figure 1  The effect of 103.45 mM NaCl on leaf photosynthesis.** (A) Photosynthetic rate ($A_n$), (B) the ratio of intercellular to ambient $CO_2$ concentration ($C_i/C_a$), (C) stomatal conductance ($g_s$), (D) intrinsic water use efficiency ($WUE_i$) in leaves of camphor seedlings at at 103.45 mM NaCl at 30d, 50d, 80d. Each panel shows data for the zero salt control (CK, grey bar) and salt-stressed control (103.45 mM NaCl, black bar). Each bar represents the mean ($\pm$SE) of three replicates. T, N and T × N are the $P$-values for the time of NaCl treatments effect, the NaCl treatments effect, and the interaction effect between the time of NaCl treatments and the NaCl treatments, respectively. Different letters above bars show significant differences between means ($P < 0.05$).

(Fig. 3 and Table 2). Salinity induced a strong increase in malondialdehyde (MDA) at each time period (6–14 fold that of the CK), combined with greatly decreased activity of peroxidase (POD), which decreased by 9.4-fold after 30 days, 15.7-fold after 50 days and 16-fold after 80 days compared to that of the CK, and the activity of POD showed no difference at 50 days or 80 days in salinity treatment. Superoxide dismutase (SOD) activity significantly increased at 30 days and was 3.5 times that of the CK, after which it returned

**Table 2  The $p$-value in each index between NaCl stress and the time of NaCl treatment.** NaCl stress (N), the time of NaCl treatments (T), NaCl treatments and NaCl stress (T × N).

| Indexes | T | N | T*N |
|---|---|---|---|
| $A_n$ | <0.0001 | <0.0001 | <0.0001 |
| $g_s$ | 0.0220 | <0.0001 | 0.3250 |
| $C_i/C_a$ | 0.0304 | 0.0017 | 0.0087 |
| $WUE_i$ | 0.7807 | 0.0017 | 0.9982 |
| $F_m$ | 0.0095 | <0.0001 | 0.9427 |
| $F_v/F_m$ | 0.0006 | <0.0001 | 0.0004 |
| NPQ | 0.0785 | 0.0003 | 0.0279 |
| $H_d$ | 0.0110 | 0.0006 | 0.0263 |
| ΦPSII | 0.0020 | <0.0001 | 0.3763 |
| qP | 0.0176 | <0.0001 | 0.3328 |
| Chl a | 0.0564 | <0.0001 | 0.0721 |
| Chl b | 0.0176 | <0.0001 | 0.2306 |
| Chl a/Chl b | 0.9313 | 0.0128 | 0.0494 |
| Chl | 0.0126 | <0.0001 | 0.0345 |
| MDA | <0.0001 | <0.0001 | <0.0001 |
| POD | 0.0041 | <0.0001 | 0.0096 |
| SOD | <0.0001 | 0.0002 | <0.0001 |

**Note**
The bold styling of values signifies that the P–value is less than 0.05.

to a normal level after 50 days and 80 days. The ANOVA analyses showed the difference of MDA, POD, SOD among T, N and T × N with $P < 0.01$ (Fig. 4 and Table 2).

The ultrastructure of the chloroplasts at 50 days in response to NaCl treatment is shown in Fig. 5, which illustrates greater number of chloroplasts in the CK samples than in salinity stress treated samples; the chloroplasts had much more space between each other, and most chloroplasts were far apart from the plasma membrane (PM) under salt stress (Figs. 5A and 5D). Without any salt stress treatment, the chloroplasts were close to the cell wall (CW) and had a more integrated structure, a larger shape of starch granules (SG), smaller numbers of osmium granules (OS), and tighter and more regular lamellar structure of thylakoids (Thl) than did the control chloroplasts (Figs. 5B and 5C). Salinity caused severe damage to chloroplasts, which was reflected by the injured chloroplast membrane (ChM), swollen SG and the loose and irregular lamellar structure of the thylakoids (Thl) (Figs. 5E and 5F).

## Expressional and functional analysis of differential proteins

There were 2,291 proteins certificated and 240 differential expression proteins (DEPs) in the salinity compared with group (S/C) (Table 3), including 95 up-regulated proteins and 145 down-regulated proteins The 62 upregulated proteins were involved in 17 pathways and 33 up-regulated proteins were unknown, while 106 down-expressed DEPs were involved in 18 pathways (Table S1).

The results of the analysis of the biological process, cellular component and molecular function were arranged to the significance determined by $p$-value (Figs. 6A, 6B and 6C).

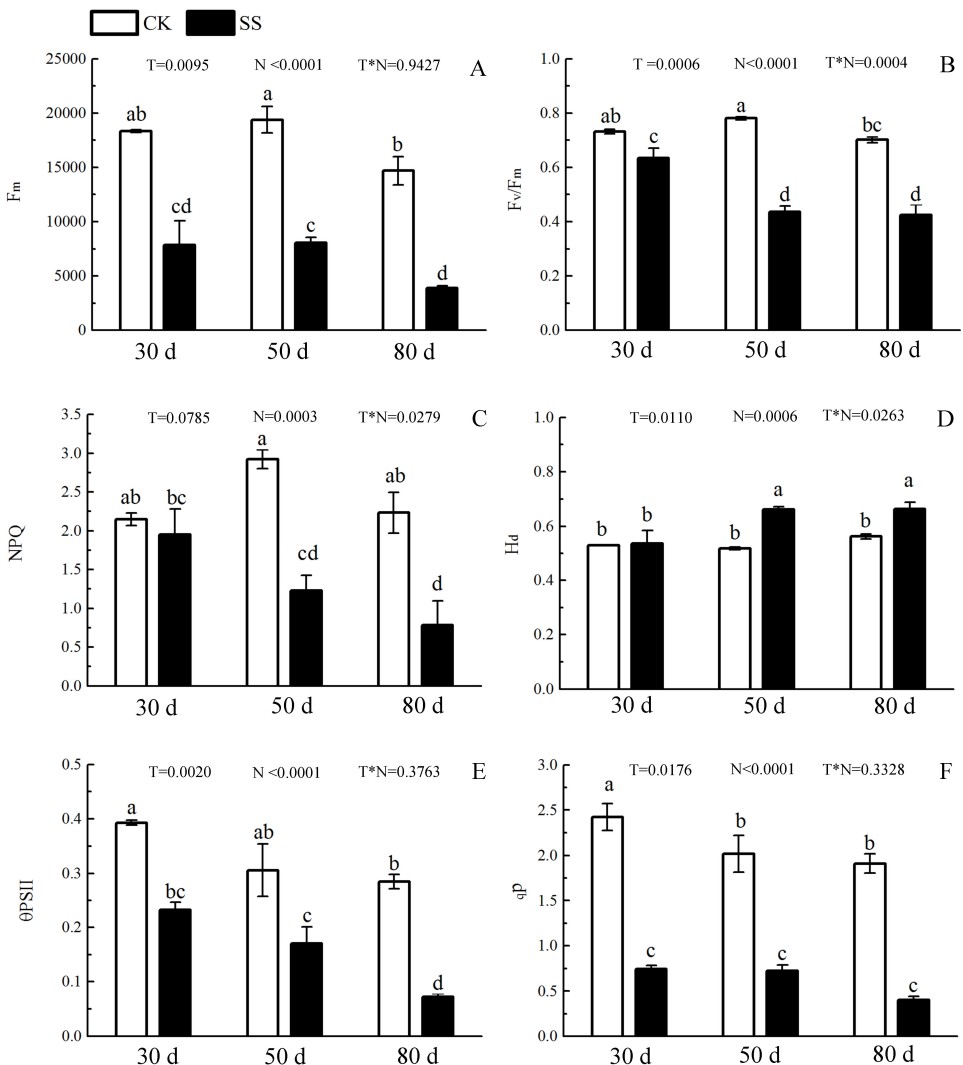

**Figure 2** **The effect of 103.45 mM NaCl on Chlorophyllfluorescence parameters.** (A) Maximal chlorophyll fluorescence emission ($F_m$), (B) maximum quantum yield of PSII ($F_v/F_m$), (C) non-photochemical quenching (NPQ), (D) thermal dissipation ($H_d$), (E) relative quantum efficiency of PSII photochemistry (ΦPSII) and (F) photochemical quenching coefficient (qP) of leaves of camphor seedlings. Other details as in Fig. 1.

All of the functions were clearly related to those DEPs ($P < 0.01$). The most obvious DEPs affecting the biological processes were the "generation of precursor metabolites and energy" and "single-organism metabolic process". In the cell component category, the most significantly influenced components by salinity were chloroplast and cytoplasm. The results of molecular function category showed that the salinity influenced the transformation of energy, including the NAD binding, donors of NAD or NADP. Other effects on processes were related to ATPase activity, the electron transport pathway of photosynthesis activity, and chlorophyll binding.

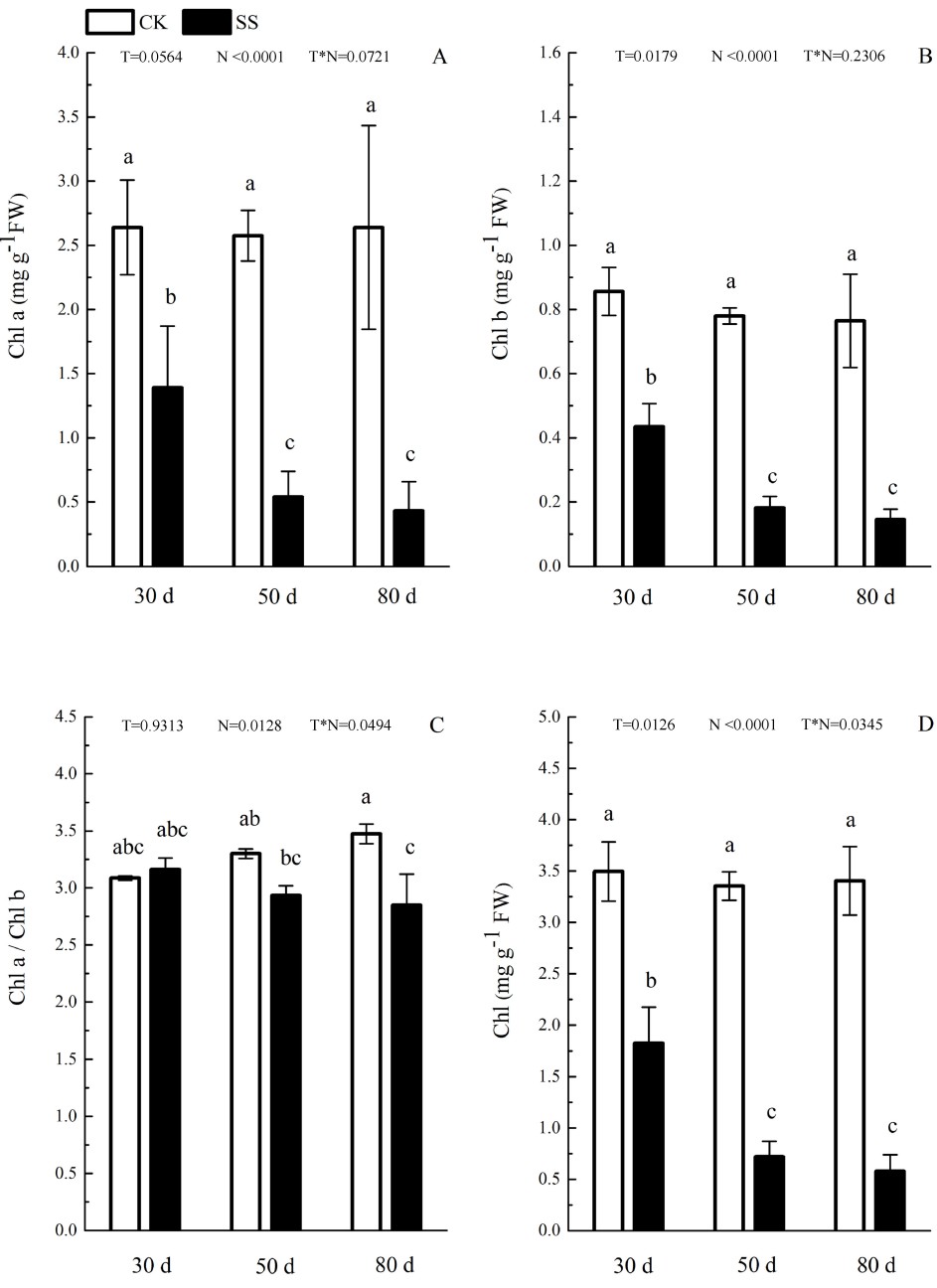

**Figure 3** **The effect of103.45 mM NaCl on leaves pigments.** (A) Chlorophyll a (Chl a) and (B) chlorophyll b (Chl b), (C) ratio of Chl a and Chl b (Chl a/Chl b), (D) total chlorophyll (Chl) of camphor leaves. Other details as in Fig. 1.

## Functions classification of and photosynthesis pathway effected by DEPs in response to salt stress

Based on KEGG database analysis, over half of DEPs function was either not clear (40.97%) or involved other less effected KEGG pathways (10.57%). The specific pathway affected by most DEPs was the metabolic pathway (24.23%) (Fig. 7A). Although photosynthesis was

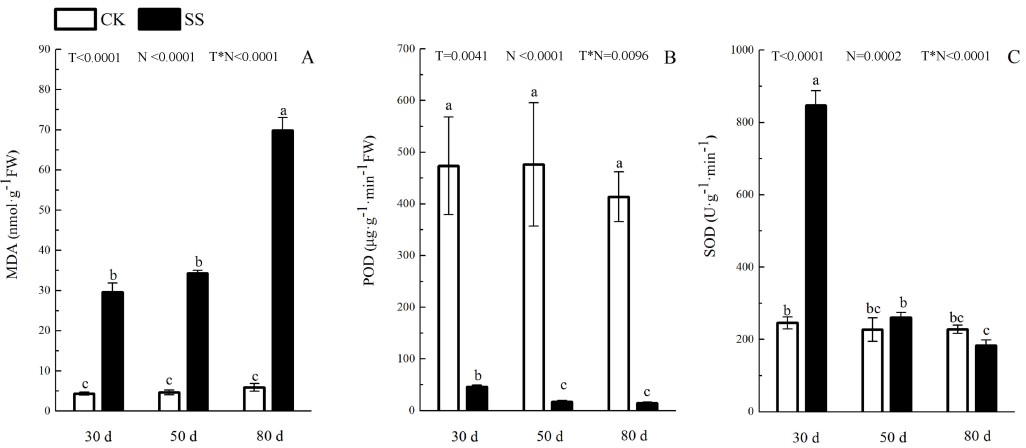

**Figure 4  The effect of 103.45 mM NaCl on leaves malondialdehyde and antioxidant enzyme.** (A) The amount of malondialdehyde (MDA), (B) the activity of Peroxidase (POD) and (C) the activity of superoxide dismutase (SOD) of leaves of camphor seedlings. Other details as in Fig. 1.

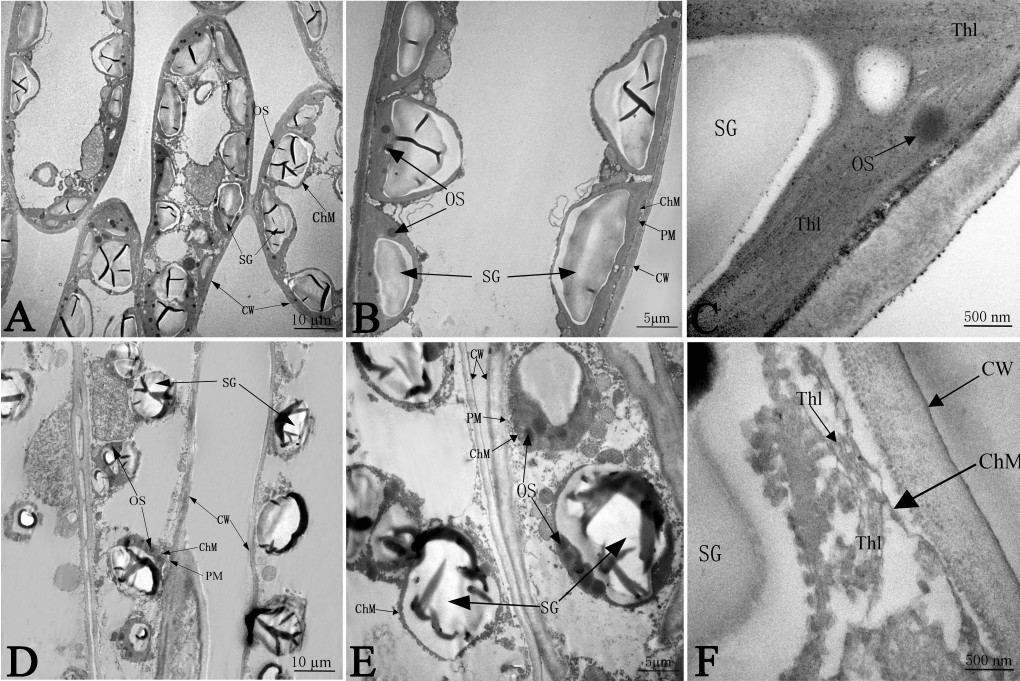

**Figure 5  The effect of 103.45 mM NaCl on leaves chloroplast ultrastructure of camphor seedlings.** Non-salinity treatment (A, B, C) and 103.45 mM NaCl treatment (D, E, F). CW, cell wall; PM, plasma membrane; ChM, chloroplast membrane; SG, starch granule; OS, osmiophilic plastoglobuli; Thl, lamellar structure of thylakoids. Scale bars are 10 µm, 5 µm and 500 nm.

**Table 3  Numbers ofidentified proteins in leaves of Campho rseedling (*Cinnamomum camphora* L.).** Definitions as follow: no salinity treatment (C), 103.45 mM NaCl treatment (S).

| Total number of all proteins | Total number of reliable proteins | Number of differential proteins in Dynamics | The number of differential proteins in S/C |
|---|---|---|---|
| 2,291 | 1,423 | 666 | 240 |

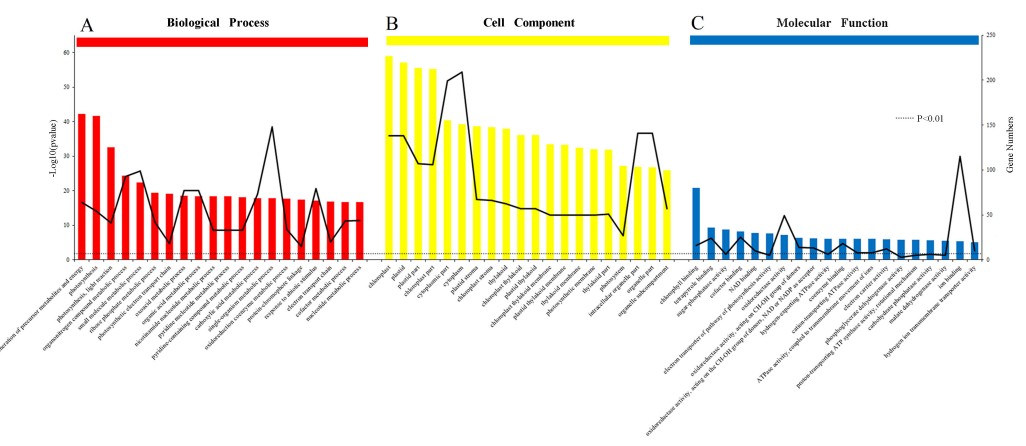

**Figure 6  Gene Ontology (GO) classification of proteins incamphor leaves.** Comparison of differentially expressed proteins (DEPs) under 103.45 mM NaCl stress responses. (A) The biological process, (B) the cell component and (C) the molecular function. The bar: the significance of each foundation enriched by DEPs and the black fold line: the number of DEPs enriched on each foundation. The black dotted line: the significant standard $P$-value $< 0.01$.

the second pathway affected in terms of the amount of DEPs (11.45%), photosynthesis was still the most significant influenced function ($P < 0.01$) (Figs. 7A & 7B). Then, 31 DEPs (1 upregulated and 30 downregulated expression) positively played the role in the photosynthetic process whose cellular locations were chloroplasts. (Figs. 8A & 8B).

The only increased DEP was PSBS, named photosystem II 22 kDa family protein. Its molecular function and biological process were chlorophyll binding and photosynthesis respectively, and it was located on PsbS site. Eight proteins were targeted to the sites of PsbA, PsbD, PsbC, PsbB, PsbE, PsbO, PsbP, PsbQ in the photosystem II, which were PsbA, PsbD, PsbC, PsbB, PsbE, PSBP1, PSBO2 and PSBQ2, respectively. Seven DEPs joining photosystem I were identified: psaB, psaC, PSAD2, PSAN, PSAE2, PSAH2, PSAL. The DEPs (PetB and PetD) targeting the petB and petD sites play a role in the Cytochrome b6/f complex. Several DEPs, atpB, atpA, ATPC1 and F10M6.100, affected the F-type ATPase which affected the beta, alpha, gamma and b sites. In our research, we could not find any DEPs that played a role in photosynthetic electron transport (Fig. 8A and Table S2). NaCl also affected the LHC, as 9 downregulated DEPs named Lhca1-Lhca4 and Lhcb1-Lhcb5 were identified (Fig. 8B). In this study, 31 DEPs participated in biological processes, including photosynthesis (20 DEPs), the purine ribonucleoside monophosphate metabolic process (atpB), the organonitrogen compound metabolic process (atpA), primary metabolic processes (4 DEPs), the response to abiotic stimulus (2 DEPs), the

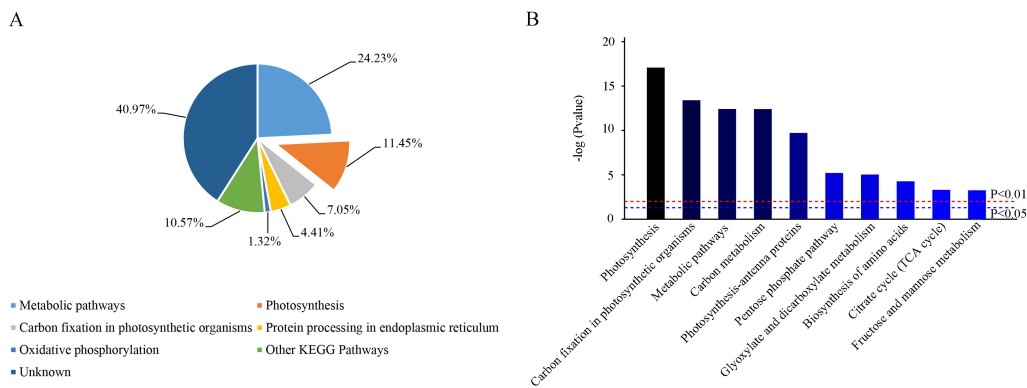

**Figure 7** **Functional classification of identified proteins of camphor leaves in response to 103.45 mM NaCl stress.** DEPs were catalogued according to the Kyoto Encyclopedia of Genes and Genome (KEGG) database. (A) The percentage for each class is shown as represented in the pie-chart. (B) The column was the significance of different functions. The red and blue dot line were significant standard: $P < 0.01$ and $P < 0.05$.

generation of precursor metabolites and energy (LHCB2.1), the regulation of phosphate metabolic process (LHCB4.2), small molecule metabolic processes (LHCB5) and carboxylic acid metabolic processes (T25B24.12). All of these DEPs were strongly associated with chlorophyll binding (16 DEPs) and other functions, including purine ribonucleoside triphosphate binding, hydrogen-exporting ATPase activity, tetrapyrrole binding, electron transport, oxidoreductase activity, binding, ion binding and hydrogen ion transmembrane transporter activity (Table S2).

## DEPs positively affected the oxidative process by GO analysis

According to GO analysis, there were 17 biological processes related to oxidative stress by 106 DEPs, and much of the DEPs related were downregulated (Table S3). The oxidoreduction coenzyme metabolic process, response to oxidative stress, response to hydrogen peroxide, hydrogen peroxide metabolic process and oxidation–reduction process were affected by most DEPs (Fig. 9). There were 57 DEPs that were highly involved in the redox effect because of their involvement in at least two oxidative functions (Table S4 and Table 4). From the expression of proteins, the highest- and lowest-expressed DEPs were nitronate monooxygenase protein (10.33 fold) (gi|743899657) and 18.2 kDa class I heat shock family protein (0.09 fold) (gi|566175391). The 9 DEPs were certificated and named heat shock family protein, $O_2$ evolving complex family protein, malate dehydrogenase family protein, Mitogen-activated protein and anti-oxidative protein. Moreover, there were 6 redox DEPs that directly affected photosynthesis via the same gene (Table 4 and Table S2). These proteins were the $O_2$ evolving complex 33kD family protein (gi|224094610) and 5 predicted proteins: oxygen-evolving enhancer protein 1 (gi|743921083), oxygen-evolving enhancer protein 2 (gi|743840443), ATP synthase gamma chain, (gi|743885735), cytochrome b6 (gi|743810316); and ATP synthase subunit b' (gi|743881832). All those directly effective DEPs were downregulated, which could explain the inhibition of photosynthesis caused by 103.45 mM NaCl. (Table 4).

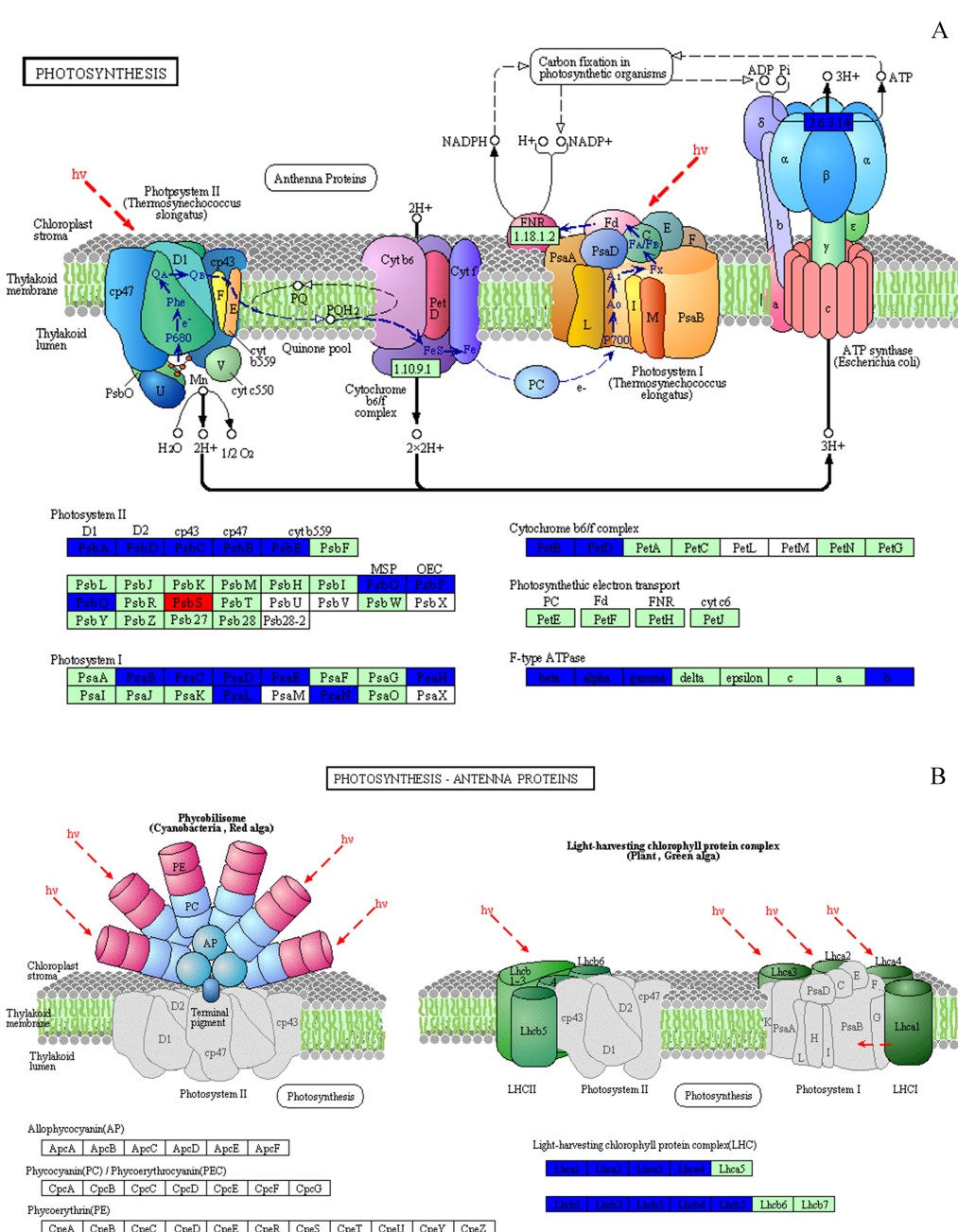

**Figure 8 Effect of 103.45 mM NaCl on DEPs enriched onphotosynthetic process.** (A) Photosynthesis pathway and (B) photosynthesis antenna proteins. Red (increase) or blue (decrease) boxes indicated DEPs compared non-salinity and 103.45 mM NaCl treatment. The basal figure cited from the KEGG PATHWAY Database (https://www.kegg.jp/kegg-bin/show_pathway?map=map00195&show_ description=show). Definition of proteins: (1) PsbA, PSII P680 reaction center D1 protein; (2) PsbB, PSII CP47 chlorophyll apoprotein; (3) PsbC, PSII CP43 chlorophyll apoprotein; (4) PsbD, PSII P680 reaction center D2 protein; (5) PsbE, PSII cytochrome b559 subunit alpha; (6) PsbO, PSII oxygen-evolving enhancer protein 1; (7) PsbP, PSII oxygen-evolving enhancer protein 2; (continued on next page...)

**Figure 8 (…continued)**
(8) PsbQ, PSII oxygen-evolving enhancer protein 3; (9) PsbS, PSII 22 kDa protein; (10) PsaB, PSI P700 Chl a apoprotein A2; (11) PsaC, PSI subunit VII; (12) PsaD, PSI subunit II; (13) PsaE, PSI reaction center subunit IV; (14) PsaH, PSI subunit VI; (15) PsaL, PSI subunit XI; (16) PsaN, PSI subunit PsaN; (17) PetB, cytochrome b6; (18) PetD, cytochrome b6-f complex subunit 4; (19) beta, F-type $H^+$-transporting ATPase subunit beta; (20) alpha, F-type $H^+$-transporting ATPase subunit alpha; (21) gamma, F-type $H^+$-transporting ATPase subunit gamma; (22) b, F-type $H^+$-transporting ATPase subunit b; (23) Lhca1~4, light-harvesting complex I chlorophyll a/b binding protein 1~4; (24) Lhcb1~5, light-harvesting complex II chlorophyll a/b binding protein 1~5.

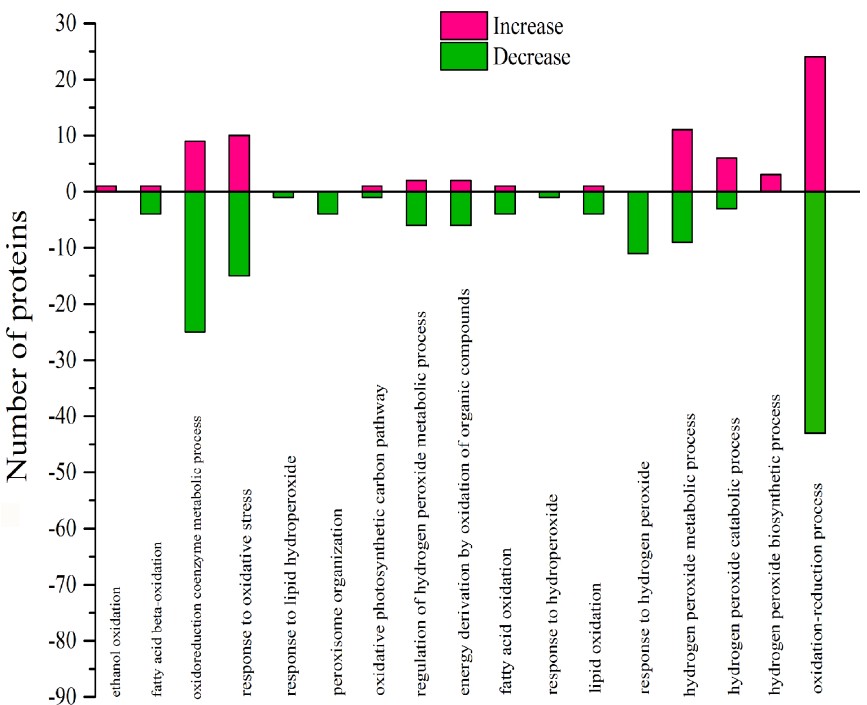

**Figure 9   Classification of DEPs enriched in oxidative functionscaused by 103.45 mM NaCl.** The pink column shows up-regulated proteins and the green one shows down-regulated proteins caused by 103.45 mM NaCl. Pink (increase) or green (decrease) columns indicated the expression of DEPs.

## DISCUSSION

Salinity led to the growth limitation of camphor seedlings with the weak photosynthetic ability. At the early stage of salt stress (30 days), stomatal limitation was the main factor for photo-inhibition, while the non-stomatal effects (e.g., disrupted chloroplasts, ion toxicity, osmotic stress) including oxidative effects (e.g., MDA, SOD, POD change) might play more important roles in photosynthetic inhibition at the late stage. Similarly, camphor seedlings exposed to $O_3$ for two years exhibited reductions in $A_n$ which verified the damage to ribulose-1,5-bisphosphate carboxylase/oxygenase and the photochemical apparatus were reason for photosynthetic limitation (*Niu et al., 2014*). The change in fluorescence index and pigments implied that salinity led the inhibitory action of salt ions on the PSII reaction centre, which could be related to the harm of the photosynthetic

**Table 4  The DEPs enriched on oxidative process.** Proteins are regard as differentially regulated proteins if the proteins abundance was equal to or greater than 2.0-fold or less than 0.5-fold.

| N | ID | Protein name | UniProt | Genename | S/C |
|---|---|---|---|---|---|
| 196 | gi\|566175391 | 18.2 kDa class I heat shock family protein | P19037 | HSP18.1 | 0.09 ± 0.04 |
| 483 | gi\|566167461 | 17.6 kDa class I small heat shock family protein | Q9ZW31 | HSP17.6B | 0.35 ± 0.11 |
| 440 | gi\|224120952 | heat shock 22K family protein | P31170 | HSP25.3 | 0.3 ± 0.02 |
| 292 | gi\|566168071 | Superoxide dismutase family protein | O78310 | CSD2 | 0.1 ± 0.02 |
| 579 | gi\|224094610 | O2 evolving complex 33kD family protein | Q9S841 | PSBO2 | 0.23 ± 0.17 |
| 1028 | gi\|224056321 | malate dehydrogenase family protein | O82399 | At2g22780 | 0.5 ± 0.04 |
| 1047 | gi\|224067920 | Mitogen-activated protein kinase 4 | Q39024 | MPK4 | 0.47 ± 0.02 |
| 184 | gi\|566179485 | alcohol-dehydrogenase family protein | Q96533 | ADH2 | 2.93 ± 1.4 |
| 1523 | gi\|591403292 | class III peroxidase | Q9LE15 | PER4 | 2.17 ± 0.6 |
| 41 | gi\|743921083 | PREDICTED: oxygen-evolving enhancer protein 1, chloroplastic | Q9S841 | PSBO2 | 0.12 ± 0.01 |
| 42 | gi\|743875087 | PREDICTED: glyceraldehyde-3-phosphate dehydrogenase B, chloroplastic isoform X1 | P25857 | GAPB | 0.32 ± 0.03 |
| 48 | gi\|743924688 | PREDICTED: presequence protease 1, chloroplastic/mitochondrial | Q9LJL3 | PREP1 | 2.3 ± 0.09 |
| 58 | gi\|743926242 | PREDICTED: chaperone protein ClpB1 | P42730 | CLPB1 | 0.23 ± 0.01 |
| 66 | gi\|743906507 | PREDICTED: glycerate dehydrogenase | Q9C9W5 | HPR | 0.4 ± 0.04 |
| 89 | gi\|743879686 | PREDICTED: sedoheptulose-1,7-bisphosphatase, chloroplastic | P46283 | At3g55800 | 0.19 ± 0.05 |
| 93 | gi\|743885735 | PREDICTED: ATP synthase gamma chain, chloroplastic | Q01908 | ATPC1 | 0.41 ± 0.07 |
| 95 | gi\|743895899 | PREDICTED: catalase isozyme 1 | P25819 | CAT2 | 0.38 ± 0.06 |
| 153 | gi\|743840443 | PREDICTED: oxygen-evolving enhancer protein 2, chloroplastic-like | Q42029 | PSBP1 | 0.12 ± 0.01 |
| 176 | gi\|743822923 | PREDICTED: malate dehydrogenase, glyoxysomal | O82399 | At2g22780 | 0.22 ± 0.1 |
| 205 | gi\|743889355 | PREDICTED: probable fructose-bisphosphate aldolase 3, chloroplastic | Q9ZU52 | FBA3 | 3.25 ± 1.48 |
| 210 | gi\|743819545 | PREDICTED: uncharacterized protein LOC105123107 | Q9LYR4 | T22N19_70 | 2.03 ± 0.56 |
| 224 | gi\|743924958 | PREDICTED: pyruvate dehydrogenase E1 component subunit alpha, mitochondrial-like | P52901 | E1 ALPHA | 3.1 ± 0.27 |
| 240 | gi\|743916626 | PREDICTED: citrate synthase, mitochondrial isoform X4 | P20115 | CSY4 | 3.61 ± 0.77 |
| 253 | gi\|743810316 | PREDICTED: cytochrome b6 | P56773 | petB | 0.19 ± 0.04 |
| 297 | gi\|743795553 | PREDICTED: malate dehydrogenase, glyoxysomal | O82399 | At2g22780 | 0.2 ± 0.05 |
| 334 | gi\|743844327 | PREDICTED: probable mediator of RNA polymerase II transcription subunit 37c isoform X2 | Q9LHA8 | MED37C | 0.31 ± 0.05 |
| 402 | gi\|743912326 | PREDICTED: probable aldo-keto reductase 2 | Q93ZN2 | At1g60710 | 2.11 ± 0.71 |
| 421 | gi\|743812897 | PREDICTED: 17.9 kDa class II heat shock protein-like | P29830 | HSP17.6 | 0.13 ± 0.01 |
| 449 | gi\|743811825 | PREDICTED: probable pyridoxal biosynthesis protein PDX1 | Q8L940 | PDX13 | 0.43 ± 0.05 |
| 450 | gi\|743899195 | PREDICTED: peroxidase 12-like | Q96520 | PER12 | 4.27 ± 0.28 |
| 452 | gi\|743907790 | PREDICTED: isocitrate dehydrogenase NADP | Q8LPJ5 | At5g14590 | 0.47 ± 0.07 |
| 493 | gi\|743928577 | PREDICTED: uncharacterized protein LOC105113849 | Q9FX85 | PER10 | 0.46 ± 0.1 |
| 499 | gi\|743874087 | PREDICTED: heat shock 70 kDa protein isoform X2 | Q9LHA8 | MED37C | 0.33 ± 0.04 |

**Table 4** (*continued*)

| N | ID | Protein name | UniProt | Genename | S/C |
|---|---|---|---|---|---|
| 506 | gi\|743937119 | PREDICTED: fructose-1,6-bisphosphatase, chloroplastic-like | P25851 | FBP | 0.46 ± 0.01 |
| 515 | gi\|743879261 | PREDICTED: ferritin-3, chloroplastic-like | Q9SRL5 | FER2 | 4.83 ± 0.7 |
| 596 | gi\|743814108 | PREDICTED: aspartate aminotransferase, cytoplasmic | P46644 | ASP3 | 3.17 ± 1.13 |
| 703 | gi\|743832246 | PREDICTED: vacuolar protein sorting-associated protein 4B-like | Q9ZNT0 | SKD1 | 0.31 ± 0.11 |
| 813 | gi\|743796298 | PREDICTED: uncharacterized protein LOC105111212 | O24520 | AHB1 | 2.13 ± 0.11 |
| 866 | gi\|743910930 | PREDICTED: pyruvate dehydrogenase (acetyl-transferring) kinase, mitochondrial | P49333 | ETR1 | 2.65 ± 1.44 |
| 931 | gi\|743794407 | PREDICTED: presequence protease 1, chloroplastic/mitochondrial isoform X2 | Q9LJL3 | PREP1 | 2.46 ± 0.13 |
| 962 | gi\|743852770 | PREDICTED: monodehydroascorbate reductase, chloroplast | P92947 | At1g63940 | 2 ± 0.1 |
| 996 | gi\|743881832 | PREDICTED: ATP synthase subunit b', chloroplastic-like | Q42139 | F10M6.100 | 0.4 ± 0.22 |
| 1213 | gi\|743932656 | PREDICTED: hevamine-A-like | P19172 | CHIB1 | 3.17 ± 0.35 |
| 1401 | gi\|743939144 | PREDICTED: uncharacterized protein LOC105117914 | Q8RXS1 | PNSB4 | 0.36 ± 0.01 |
| 1473 | gi\|743899657 | PREDICTED: nitronate monooxygenase | Q24JJ8 | GLO3 | 10.33 ± 7.96 |
| 1534 | gi\|743928583 | PREDICTED: NAD(P)H-quinone oxidoreductase subunit N, organellar chromatophore | Q9LVM2 | ndhN | 0.34 ± 0.14 |
| 413 | gi\|566151722 | hypothetical protein POPTR_0001s29700g | Q9LMU1 | EGY3 | 0.33 ± 0.21 |
| 223 | gi\|566187985 | hypothetical protein POPTR_0009s14500g | Q1H537 | DVR | 0.48 ± 0.04 |
| 386 | gi\|566189392 | hypothetical protein POPTR_0010s06320g | P31170 | HSP25.3 | 0.33 ± 0.09 |
| 389 | gi\|566190935 | hypothetical protein POPTR_0010s15200g | A8MS68 | LPD1 | 2.84 ± 1.48 |
| 620 | gi\|566165056 | hypothetical protein POPTR_0004s05340g | Q9ZP06 | At1g53240 | 0.28 ± 0.05 |
| 817 | gi\|566206271 | hypothetical protein POPTR_0015s06770g | P42730 | CLPB1 | 0.29 ± 0.05 |
| 1034 | gi\|566148978 | hypothetical protein POPTR_0001s14630g | Q9M9E1 | ABCG40 | 2.15 ± 1.24 |
| 1055 | gi\|566176419 | hypothetical protein POPTR_0006s14520g | Q9FUP0 | OPR3 | 3.06 ± 0.49 |
| 1098 | gi\|566209778 | hypothetical protein POPTR_0016s11621g | P25851 | FBP | 0.39 ± 0.1 |
| 1499 | gi\|566186111 | hypothetical protein POPTR_0009s03520g | P13853 | HSP17.6C | 0.13 ± 0.05 |
| 1541 | gi\|566167975 | hypothetical protein POPTR_0004s22180g | Q9FUP0 | OPR3 | 4.02 ± 1.13 |

organs (*Wang et al., 2007*; *Yue et al., 2019*), especially at 50 days of NaCl treatment in our research, similar results in cucumber seedlings (*Cucumis sativus* cv. Jingyou No. 4) under 75 mM NaCl (*Shu et al., 2012*) and in dianthus superbus (*Caryophyllaceae*) under 155.18 mM NaCl (*Ma et al., 2017*). Then, chloroplasts take apart in many important metabolic processes, and approximately 300 distinct proteins have been assessed to be located within chloroplasts, which are almost role in photosynthetic process (*Suo et al., 2017*).

## NaCl stress affected proteins involved in photosynthetic process

During photosynthesis, the light reactions are mainly driven by four intrinsic multi-subunit membrane-protein complexes (PSI, PSII, the cytochrome b6/f complex, ATP synthase) (*Nelson & Yocum, 2006*). In this study, the only 1 upregulated DEP, Photosystem II 22 kDa family protein (PsbS), caused by NaCl stress, affects chlorophyll binding. This protein,

located in thylakoid membranes, plays a role in NPQ by regulating the interaction between LHCII and PS II in the granum membranes (*Kiss, Ruban & Horton, 2008*). Zhang and Yang showed that PsbS gene expression in a non-hyperaccumulating ecotype (NHE) of *sedum alfredii* was more than 2 times higher than that in control plants under Cd stress and was accompanied by a significant loss in PSII photochemical efficiency and reduced NPQ values (*Zhang & Yang, 2014*). Semblable results appeared in our research that the expression of PsbS was also more than 2 times, accompanied by a 42.1% reduction in NPQ at 50 days. The rest of other proteins in photosystem II (PSII) were downregulated, including psbA, psbD, psbC, psbB, PSAE2, PSBO2, PSBP1, and PSBQ2. The changes in above proteins was consilient with cucumber research under salinity stress for 7 days to show 23 downregulated proteins, including psbA, ATP synthase beta subunit, CP47 and so on (*Shu et al., 2015*). The proteins of psbA, psbD and cyt b559 form the PS II reaction center complex together (*Van Wijk et al., 1997*), which could be sensitive to the environment. In our study, the downregulation of the cyt b559 (PsbE) was induced by 103.45 mM NaCl stress at 50 days, which indicated that the camphor seedlings were sensitive to 103.45 mM NaCl. Similar results were found in the research of salt-tolerance in a Mediterranean sea grass species (*Cymodocea nodosa*) under hypersaline exposure to show the increased accumulation of cyt b559 (*Piro et al., 2015*) with no alteration of the maximum efficiency of PS II ($F_v/F_m$). Different samples caused the difference in proteins expression between our results and research on cotton (*Gossypium hirsutum* L) under salt stress. Salinity caused down-expression of proteins located on PS II (PsbA-E) and PS I (PsaA, PsaB, PsaF, PsaG, PsaL and PsaN), PetA and PetD in cytochrome b6/f complex, and F-type $H^+$-transporting ATPase subunit b on cotton, which suggests that cotton is sensitive to 240 mM NaCl (*Chen et al., 2016*). Our results also showed the downregulation of PSII (PsbA-E, PsbO-Q) and PSI (PsaB-E, PsaH, PsaL and PsaN), of PetB, PetD in cytochrome b6/f complex, and of ATP synthase subunit b' (F10M6.100) which is involved in hydrogen ion transmembrane transporter activity and indicated an obviously negative effect of 103.45 mM NaCl on camphor seedlings (Fig. 8 and Table S2). It has been reported that the upregulation of some above proteins can provide protection from stress during photosynthesis. For example, the polyploidy of autotetraploid *Paulownia tomentosa* (PT4) was more vigorous, adaptive, and capable of surviving better than diploid *P. tomentosa* (PT2) in harsher environment because of some upregulated DEPs such as photosystem I reaction center subunit IV B (PsaE-2), and putative cytochrome b6f Rieske iron-sulfur subunit (*Yan et al., 2017*); heat stress recovery caused a 5.1-fold expression of PsaH and a 28.1-fold increase in expression of PsaN to get more temperature resistance of grapevine (*Liu et al., 2014*). Therefore, it inferred salinity induced the photo-inhibition in camphor seedlings by the downregulation of proteins in PSI, PSII, the cytochrome b6/f complex, ATP synthase without a protective role.

PSII and PSI are closely linked to light-driven electron transport and play a major role in the different light absorption properties of peripheral antennae, which are formed within light-harvesting chlorophyll protein complexes (LHCs) involving LHCII and LHCI (*Pan et al., 2018*). Dalal and Tripathy verified that the reduced light absorption by antennae was the reason for the decrease in the electron transport rates of PSII and PSI (*Dalal &*

*Tripathy, 2018*). In the present study, 103.45 mM NaCl caused decreased expression of 9 DEPs within LHCs, including Lhca1-4 and Lhcb1-5, and was accompanied by reductions in $F_m$, $F_v/F_m$, NPQ, $\Phi_{PSII}$ and qP. The photosynthesis process consists of both light reactions and dark reactions. In this study, even if the integrated process or the DEPs involved in the dark reactions were not involved, the synthesized energy (ATP, NADPH) applied to the dark reactions (*Muench, Trinick & Harrison, 2011*) was inhibited by salinity. From research on stress resistance in black locust (*Robinia pseudoacacia* L), it has been inferred that a 2 × locust plant had weakened resistance to salinity than a 4 × locust plant because of the decrease in ATP synthase CF1 beta subunit (*Meng et al., 2016*). Downregulated expression of atpB (beta), atpA (alpha), ATPC1 (gamma) and F10M6.100 (b) of the F-type ATPase (fold-changes of 0.2, 0.23, 0.41 and 0.4) were observed in salt-stressed camphor seedlings. These results indicated that ATPase synthase was severely disrupted by 103.45 mM NaCl, which reduced the resistance to salinity of camphor seedlings by weakening the dark reactions in photosynthesis. The research of *brassica napus* leaves reported photo-inhibition under 200 mM NaCl stress with the similar reason by the significantly downregulated expression of proteins related to ATPase synthase, including ATP synthase CF1 beta subunit, ATP synthase gamma chain, F1-ATPase alpha subunit, ATP synthase CF1 alpha subunit, and ATP synthase beta subunit (*Jia et al., 2015*).

## Photosynthesis limitation due to oxidative effect caused by NaCl stress

There were 17 biological processes that were related to the oxidative functions and that were especially associated with 57 highly effective DEPs, which joined at least two oxidative functions. The DEPs with oxidative function occupied 23.75% of the total DEPs in S/C. As the same gene name appeared in photosynthesis and functions related to redox process, it was concluded that 6 proteins in photo inhibition caused by salinity were highly effect: the O2 evolving complex 33kD family protein (PSBO2, 0.23 ± 0.17), oxygen-evolving enhancer protein 1 (PSBO2, 0.12 ± 0.01), oxygen-evolving enhancer protein 2 (PSBP1, 0.12 ± 0.01), ATP synthase gamma chain (ATPC1, 0.41 ± 0.07), cytochrome b6 (petB, 0.19 ± 0.04), and ATP synthase subunit b' (F10M6.100, 0.4 ± 0.22) (Table.4 & Table.S2). Besides, there were 9 verified DEPs of 57 highly affected DEPs: 2 were antioxidative proteins, 3 belonged to the heat-shock family and the others were O2-evolving complex 33 kD family proteins, malate dehydrogenase family proteins, mitogen-activated protein kinase 4 and alcohol dehydrogenase family proteins. The 103.45 mM NaCl induced decreased POD activity and increased SOD activity, but the expression of superoxide dismutase family proteins were downregulated, and the expression of class III peroxidase was weakly elevated. These results could be the reason for the detection of the antioxidant enzymes from the reduction of antioxidant isoenzymes related to the content of ROS. Similar results were verified in a study on the effects of NaCl on *Broussonetia papyrifera* to show the different activities of SOD, POD, and CAT in leaves, stems, and roots accompanying up/down-regulated expression of SOD, POD, and CAT isoenzymes caused by 50, 100, and 150 mM NaCl treatments (*Zhang et al., 2013*). Heat-shock family proteins such as sHsp, Hsp60, Hsp70, Hsp90, Hsp100 and HSF can regulate hormones, kinases, the cell cycle, the redox state,

antioxidant activity and osmolytes for stress tolerance (*Wang et al., 2004*) and the study reported that two salt-responsive heat-shock proteins (Hsp60 and Hsp70) were identified in the roots of the halophyte *Cakile maritima* under 100 and 300 mM NaCl treatments (*Belghith et al., 2018*). The O2-evolving complex 33 kD family protein in our research was downregulated (0.23 ± 0.17) and affected PSII. Similar results occurred in poplar (*Populus simonii*) research under chilling stress, which resulted in the identification of 1085 differentially expressed genes involved in photosynthesis, including one downregulated gene similar to the O2-evolving complex 33 kD family protein (-4.82-fold expression), and it affected PSII (*Song et al., 2013*). Malate dehydrogenase family proteins participate in the TCA cycle, which is the second step of respiration. A 100 mM NaCl treatment caused a 0.47-fold increased expression of malate dehydrogenase proteins in radish (*Raphanus sativus* L.) (*Sun et al., 2017*). Mitogen-activated protein kinase 4 (MKK4), a member of the MAP kinase family, is expressed in response to cellular stress (*Cuenda, 2000*). In 2012, Kim and his team verified brassinosteroid regulated stomatal development by affecting the MAPK pathway by regulating the expression of MKK4/5/7/9 (*Kim et al., 2012*), which could help regulate stress resistance in plants. Moreover, alcohol dehydrogenase is a key enzyme that catalyses the reduction of acetaldehyde to ethanol (*Chang & Meyerowitz, 1986*). In recent research, the alcohol dehydrogenase 1 has been reported in regulating the salinity stress resistance in *Arabidopsis* by overexpressing AtADH1gene (*Shi et al., 2017*). So the DEPs joined in oxidative effect could regulate the photosynthetic process to change the plant salinity tolerance.

## CONCLUSIONS

Increasing soil salinity can pose risks for camphor seedling growth because of photosynthesis inhibition, which affects the spread and usage of camphor. There were 31 proteins related to photosynthesis affected by salt stress in camphor seedlings. The results suggest that Chl biosynthesis, ATP synthesis, electron transport and transfer, oxidoreductase activity, ion transmembrane transport and chlorophyll binding in the photosynthetic process were negatively affected by 103.45 mM NaCl stress. Furthermore, this study concluded that there were 6 proteins and other verified DEPs with oxidative function, which could be the main reason explaining the photosynthetic inhibition caused by salinity. The above results enrich our knowledge of the response mechanisms of the photosynthesis process on camphor seedlings and the limiting factors caused by NaCl stress, which will better promote molecular-based breeding of salt-regulation of camphor seedlings in the future.

## ACKNOWLEDGEMENTS

We would thank Prof. Robert D. Guy, from Department of Forest and Conservation Sciences, Faculty of Forestry, University of British Columbia and Prof. Donald L. DeAngelis, from the Wetland and Aquatic Research Center, U.S. Geological Survey for their valuable comments, suggestions and language modification that have greatly improved the quality of this manuscript.

### Funding

This work was supported by the Jiangsu Agriculture Science and Technology Innovation Fund (Grant No. CX(17)1004), the National Special Fund for Forestry Scientific Research in the Public Interest (Grant No. 201504406), the Major Fund for Natural Science of Jiangsu Higher Education Institutions (Grant No. 15KJA220004), the National Foundation of Forestry Science and Technology Popularization (Grant No. [2015]17), the Priority Academic Program Development of Jiangsu Higher Education Institutions (PAPD), the Ningxia Autonomous Region Provincial Fund (2020AAC03091), and the Doctorate Fellowship Foundation of Nanjing Forestry University. The funders had no role in study design, data collection and analysis, decision to publish, or preparation of the manuscript.

### Grant Disclosures

The following grant information was disclosed by the authors:
Jiangsu Agriculture Science and Technology Innovation Fund: CX(17)1004.
National Special Fund for Forestry Scientific Research in the Public Interest: 201504406.
Major Fund for Natural Science of Jiangsu Higher Education Institutions: 15KJA220004.
National Foundation of Forestry Science and Technology Popularization: [2015]17.
Priority Academic Program Development of Jiangsu Higher Education Institutions (PAPD), Ningxia Autonomous Region Provincial Fund: 2020AAC03091.
Doctorate Fellowship Foundation of Nanjing Forestry University.

### Competing Interests

The authors declare there are no competing interests.

### Author Contributions

- Jiammin Yue conceived and designed the experiments, performed the experiments, analyzed the data, prepared figures and/or tables, authored or reviewed drafts of the paper, and approved the final draft.
- Dawei Shi performed the experiments, analyzed the data, authored or reviewed drafts of the paper, and approved the final draft.
- Liang Zhang and Zhiyuan Fu conceived and designed the experiments, performed the experiments, prepared figures and/or tables, and approved the final draft.
- Zihan Zhang analyzed the data, authored or reviewed drafts of the paper, and approved the final draft.
- Qiong Ren performed the experiments, authored or reviewed drafts of the paper, and approved the final draft.
- Jinchi Zhang conceived and designed the experiments, authored or reviewed drafts of the paper, and approved the final draft.

### Data Availability

    The data are available in the Supplementary Files.
## Supplemental Information

Supplemental information for this article can be found online at http://dx.doi.org/10.7717/peerj.9443#supplemental-information.

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
