# Peer review of "The photo-inhibition of camphor leaves (Cinnamomum camphora L.) by NaCl stress based on physiological, chloroplast structure and comparative proteomic analysis"

_PeerJ, doi:10.7717/peerj.9443_

## Round 0.1 · original submission · Major Revisions

Dear Dr.:

There are certainly some interesting results in this manuscript, but there are many substantial problems that are addressed by the detailed reviews provided by the referees. Please address all the points below.

The introduction and Discussion are certainly too narrow and need to address what is the main hypothesis of this study and authors should highlight and explain in deep what are their main findings in term of mechanistic approach. Furthermore references should be actualized. Reviewer 2 and 3 mentioned some in particular that should be included and discussed. And overall grammar and language style should be improved in deep.

All Reviewers had several good suggestions about material and methods and statistical presentation. The major critic is relating to the low number of replicate used for the quantification of some parameters. This issue must be justified. Also statistical model should be improved mainly considering that some results, such as fluorescence, pigments and antioxidant enzyme activity, have a temporal dimension. Maybe a GLM would contribute for results presentation, however with the low number of replication is difficult the use of this statistical approximation.

Please check gas exchange results. In my opinion An values are really low for a tree species even considering that they were not adult individual and taking into account gs values obtain. Maybe authors did not wait enough time for photosynthesis stabilization. Please check this issue. Why these measurements were not taken together with those of fluorescence in all sampling periods?. Both are not destructive techniques and this information would be complementary.
Some fluorescence parameter are missing, please include information about real quantum efficiency of PSII. Also fluorescence light-adapted parameters were obtained at 600 μmol m-2 s-1. This light intensity is not enough to obtain a real approximation to the PSII efficiency under light conditions.

Reviewer 1 ·

Basic reporting

The English of the manuscript must be improved because it is hard to understand overall.

The abbreviaton are not consistent through the manuscript. Some abbreviaton are not specificated like Caro (page 7 line 54) or PS1 and PS2 (page 8 line 67). Maybe a table with all the abbreviaton used could be useful.

Discussion's first section (page 22 line 383) is just a description of the results followed by a list of references.

Experimental design

No comment

Validity of the findings

What does biological replicate (page 17 lines 276 and 277) exactly mean?

Three replicate for the physiological analysis are not enough to obtain robust data.

Additional comments

The research question is well defined and the methods used fited the focus of the research.
However, the manuscript is not clear. The English must be improved and some sections should be written again to make them more understable.
Furthermore, the number of replicates seem insufficient for the variables measured.

Reviewer 2 ·

Basic reporting

Grammar and language style should be improved, as several sentence are not correct, e.g. “As a habitat of camphor seedlings, Jiangsu has about 15 km2 of its area along the coast is degraded…”(lines 86-90).

Many references deal with herbaceous species, mostly crops, not related taxonomically to .Cinnamomum camphora. I suggest to include only references on taxa of the same family or at least only on woody species.

Structure conforms to PeerJ standards.

Figures are relevant,of high quality, However, Figure 7 (A) (Photosynthesis pathway) and (B) (Photosynthesis antenna protein) are not original, and the information they bring is not relevant for the MS. I suggestion their deletion or at least citation of their source in the legend.

Raw data are supplied..

Experimental design

Original research within the scope of the journal.
The aims of the work are well defined.
THE MS includes an extensive data set, with a diverse methodology, but the number of biological replicates is low.

Validity of the findings

Data are insufficient for a statistically sound treatemnt.
Conclusions are well stated, linked to the aim of the work.

Additional comments

The manuscript entitled "The photo-inhibition of camphor leaves (Cinnamomum camphora L.) by NaCl stress based on physiological, chloroplast structure and comparative proteomic analysis (#34911) " deals with a topic of interest. The authors have done a good and timely need research work. The article is well organized, starting from the abstract till conclusion and includes an extensive data set. The methods used are diverse and of high standard. However, my major concern is related to the number of biological replicates, just three for physiological analysis (the minimum for a statistic treatment of the data!) and only two for proteomics (insufficient for statistics!). There is no mention of the number of replicates for the ultra structure analysis.
Regarding the Introduction section, many of the references cited deal with herbaceous species, mostly crops, not related taxonomically to .Cinnamomum camphora. I suggest to include only references on taxa of the same family or at least only on woody species. Information on the of wild habitat of camphor tree and more details about the risk of secondary salinization in the area of origin of the seeds would be of interest, and important for justifying the aim of the study.
In Material and Methods section I could not find the number of seedlings submitted to treatments and details about the saline treatment (watering into the pots, into trays, etc…). Why the rather unusual concentration of 103.45 mM NaCl was chosen?
Results are well organized ad clearly presented and figures are of good quality. However, Figure 7 (A) (Photosynthesis pathway) and (B) (Photosynthesis antenna protein are not original and in fact the information brought is not relevant. I suggestion their deletion or at least the citation of their source in the legend.
The Discussion should be focused olyon related or woody species (as mentioned for the Introduction) and not on crops.
Grammar and language style should be improved, as several sentence are not correct, e.g. “As a habitat of camphor seedlings, Jiangsu has about 15 km2 of its area along the coast is degraded…”(lines 86-90).

Reviewer 3 ·

Basic reporting

The English used through out the work is of good level, however improvements can be made. I advice the authors to review the grammar used in this work, or even pass it by a native speaking scientist for corrections.

The work itself is of good standard, and has a lot of merit. However the authors did not highlight their main findings in a sufficient manner (the increase of MDA under stress and connecting it with their other findings in regards of the buildup of oxidative stress and its effects on the photosynthetic pigments).

The literature referenced could be improved especially in the discussion as well as in the introduction. I would highly recommend including some work from Pr. Tim Flowers or other leading scientists in the field of salt stress (maybe Pr. Shabala or Pr. Hasegawa), especially their most recent reviews.

The article is well structured and the figures are of excellent quality. The buildup of the work and results seem to be in good terms with the set hypothesis.

Experimental design

The presented work is with good terms of the aims and scope of this journal, and its stated hypothesis is clear and concise. The investigation of the salt stress effects on Cinnamomum camphora, a tree of commercial and economic value to the local economy of southeast China, makes this work relevant and holds merit.

The methods used are described in a very informative and sufficient manner, and the findings are presented in a good fashion. Though some more emphasis in discussing them could elevate the quality of this work.

Validity of the findings

The findings are novel and all in all make sense with the most recent findings in the field of abiotic stress, notably salt stress in trees. My only issue is that the discussion is not thoroughly discussing and highlighting some of the findings, with a lack of cited references.

The data is robust, and seems to be of good quality, on the other hand a two-way ANOVA could have increased the resolution of the statistical differences among treatments and within every treatment at different time points, combined with a student test. However the current form of used statistics (one way ANOVA) seems to be sufficient.

This work on young plantlets (less than one year old), and as such any findings and conclusions should state this and confine any speculations to this early stage of growth in the studied tree life.

Additional comments

I would suggest addressing the comments I have stated in my submitted revision of the manuscript, as well as seeking some help from a scientific translator or a native speaking scientist. The work is of good quality and holds merit, and would be a shame to associate it with anything but an excellent level of English language.

Annotated reviews are not available for download in order to protect the identity of reviewers who chose to remain anonymous.

---

## Round 0.2 · Minor Revisions

Dear Dr. Yue,

Thanks for the intensive work done in this new version of your manuscript and to response the majority of the reviewers’ considerations. In fact, I think that this version has improved considerably compared with the previous one. However, it need a bit more work before to be completely ready for its publication in PeerJ.

There two main concerns to be considered:
Results: Although a new statistical analysis model has been used (according reviewer´ suggestions) which has certainly improved this section, I think that authors should include a Table (new one) with a summary of the main and interactive statistical effects obtained per each parameter. Also, avoid subjective terms through this section, such as “obvious”or “extremely”.
Discussion: The authors have made an important effort to improve the mechanistic approach of this section as was suggested; however, as it is stand, it is excessively long and hard to follow. Please, try to reduce this section (at least on one page), without touching the essential parts. I suggest focusing your comparisons and explanation with others studies which assessed plant responses to salinity excess, avoiding those which considered others stress factors, such as metal pollution or temperature stress.
Minor considerations:
-Please, delete Table 1 and include its information in the text.
-Please, be consistent with references through the text.
-Please, provide the specific number of replicates recorded per obtained parameter.
-Please, delete this sentence “and the tissue was obtained for 30d, 50d, and 80d salt treatments” through the text. It is redundant.

---

## Round 0.3 · Minor Revisions

Dear Dr. Yue,

Thanks for the intensive work done during the review process. However, although majority of issues have been filled in properly way. During the last checks one of our Section Editors has highlighted the follow issues.

"There are no GO terms displayed in Table 4 when the title indicates so. Though Figure 6 displays the wording for the GO: terms, it does not provide their associate GO: code term. Likewise,the deferentially expressed genes need to be listed somewhere and appropriate links created; it is suggested that it would be in Suppl. Table S1, but it is only abbreviations without any data (see lines 337-339): maybe the GO: terms can be added in S3. Some sort of validating data is needed, at least for the 240 deferentially expressed genes; better if data for all 2291 was included. The annotations for proteins are listed, but the extent of the protein identification would need presentation. There was only one condition established as salt-stress; why is there no reasoning for this singular approach? There is enough disconnection with the data that I ask for a moderate revision to build the connections of what was observed to the actual expression data with validation."

Please these aspects should be filled before manuscript will be ready for publication.

Best regards,

Enrique

---

## Round 0.4 · accepted · Accept

Dear Dr. Yue,

Thanks for the intensive work done during the review process. Just one minor suggestion, please consider to include salinity level selection justification in the Materials and Methods section, according to the information included in the rebuttal letter.

Best regards,

Enrique